# Roles of the HOX Proteins in Cancer Invasion and Metastasis

**DOI:** 10.3390/cancers13010010

**Published:** 2020-12-22

**Authors:** Ana Paço, Simone Aparecida de Bessa Garcia, Joana Leitão Castro, Ana Rita Costa-Pinto, Renata Freitas

**Affiliations:** 1BLC3—Biomassa Lenho-Celulósica de 3ª Geração, Campus of Technology and Innovation, 3405-169 Oliveira do Hospital, Portugal; 2I3S—Institute for Innovation & Health Research, University of Porto, 4200-135 Porto, Portugal; simone.bessa@i3s.up.pt (S.A.d.B.G.); joana.castro@ibmc.up.pt (J.L.C.); anap@ipatimup.pt (A.R.C.-P.); renata.freitas@ibmc.up.pt (R.F.); 3ICBAS—Institute of Biomedical Sciences Abel Salazar, University of Porto, 4050-313 Porto, Portugal

**Keywords:** *HOX*, invasion and metastasis, HMGA2/TET1/HOXA signalling pathway, TGFβ signalling, epithelial-to-mesenchymal transition, microRNAs, lncRNAs

## Abstract

**Simple Summary:**

Cancer is the second leading cause of death worldwide, right after cardiovascular diseases, and the invasion and metastatization correspond to the foremost cause of cancer-related deaths. Here, we reviewed the state of the art regarding the importance of HOX transcription factors in these last steps of cancer progression and described five of their complex mechanisms of regulation, including the miRs and lncRNAs interference. This information highlights the importance of HOX in the suppression and induction of disease advancement and point out the potential of HOX products as therapeutic targets for diverse cancer types.

**Abstract:**

Invasion and metastasis correspond to the foremost cause of cancer-related death, and the molecular networks behind these two processes are extremely complex and dependent on the intra- and extracellular conditions along with the prime of the premetastatic niche. Currently, several studies suggest an association between the levels of *HOX* genes expression and cancer cell invasion and metastasis, which favour the formation of novel tumour masses. The deregulation of *HOX* genes by HMGA2/TET1 signalling and the regulatory effect of noncoding RNAs generated by the *HOX* loci can also promote invasion and metastasis, interfering with the expression of *HOX* genes or other genes relevant to these processes. In this review, we present five molecular mechanisms of *HOX* deregulation by which the *HOX* clusters products may affect invasion and metastatic processes in solid tumours.

## 1. Introduction

The vast majority of cancer-related mortality in solid tumours is associated with the capacity of the cancer cells to invade and colonize nearby or distant vital organs forming metastasis [1], hallmarks that characterize the last steps of cancer progression [2]. A line of investigation suggests that tumour recurrence and metastization is led by a population of residual cells that survive treatment and are capable to leave their primary location. These cells disperse into the bloodstream, endure pressure in blood vessels, escape immune response, and acclimate to new cellular surroundings in a secondary site [3]. However, the development of new therapies that can eliminate residual tumour cells or prevent their action is conditioned by the incomplete understanding of the mechanisms underlying the long-term survival of these cells following treatment [4].

The metastatic process is complex and involves a multistep journey. The activation of invasion and metastasis is triggered by environmental stimuli, such as aging and circadian disruptions; adhesive signals from extracellular matrix (ECM) components, such as collagen and fibrin; ECM mechanical pressures, including tension and compression; cell–cell interactions; soluble signals, such as growth factors and cytokines; and intratumoural microbiota [5,6]. The invading tumour cells, on the way to the target site, interact with other proteins and cells to overcome the stromal challenges and complete the metastasis key steps of invasion, intravasation, circulation, extravasation and colonization as will be described throughout the text. To add more complexity to this process, secondary sites do not receive invading cancer cells passively. In fact, the host microenvironment, also designated as a premetastatic niche (PMN) or organotropic metastasis, is selectively primed by the primary tumour even before the initiation of the metastatic process. This signalling involves secretory factors and extracellular vesicles that induce vascular leakage, ECM remodelling, and immunosuppression, making the secondary microenvironment selective for the circulating tumour cells (CTCs) [7,8,9]. As common examples of organotropism, we can cite the preferred metastization of breast and prostate tumour cells to lungs, bone and liver; colon and stomach cancer to liver, lung and peritoneum; and lung carcinoma to adrenal glands and liver [10]. Hundreds of genes have been reported to lead this invasive potential, suggesting that primary tumour cells should develop a metastatic genetic signature [11]. Thus, genetic and also epigenetic modifications, accumulated during the primary tumour development and by the adaptations to the microenvironment components contribute to the metastatic process [6]. The genetic modifications involve changes in the primary DNA sequence, such as mutations, while the epigenetic mechanisms relate to chemical modifications of DNA bases and changes in the chromosomal superstructure in which DNA is packaged, such as gene promoter methylations associated with gene silencing [12,13]. In this context, alterations in *HOX* gene expression have been identified in primary tumours, metastasis and CTCs [14,15].

### 1.1. HOX Genes Genomic Organization and Transcription

*HOX* genes are a family of genes that codes for transcription factors characterized by the presence of a conserved DNA sequence designated homeobox [16]. The human genome contains 39 of those genes arranged into four clusters (*HOXA*, *HOXB*, *HOXC* and *HOXD*) located in distinct chromosomes (7p15, 17q21.2, 12q13, 2q31, respectively). Each cluster presents 9–11 genes that align in 13 paralogous groups, based on sequence homology of their homeoboxes and their position within the cluster [17]. These genes are co-ordinately transcribed following temporal and spatial collinearity with an unidirectional chromatin opening, where the chromosomic order of genes is the same as the order of its expression, forming nested and overlapping expression domains [18], that define the organogenesis along the anterior–posterior axis [19,20]. HOX proteins activity occurs with the formation of multiprotein complexes, the “hoxasomes”, composed by HOX proteins, HOX cofactors and other transcription factors [15]. These cofactors are members of the three amino acids loop extension (TALE) family, which includes PBC members (PBX1–4), HMP members (MEIS1–3 and PREP1–2), and POU family (POU1–6) [21]. The transcription of these genes happens during embryonic development but also in adult cells at lower levels [22] partaking in cellular physiology and tissue homeostasis [19,23].

### 1.2. HOX Genes and Cancer

Alterations in *HOX* gene expression have been shown in different cancer types and associated with cancer initiation and progression [15,24,25]. They can interfere with cell differentiation, survival, proliferation, angiogenesis, autophagy, inflammation and apoptosis. They also influence cell movement, migration, invasion, metalloproteinase function and regulate important stemness-related genes [26,27]. More complexly, different *HOX* genes can activate or inhibit these processes and even exhibit antagonistic behaviours in different tissues.

Thus, *HOX* genes interfere with the majority of the hallmarks of cancer, whose events must occur in parallel and/or sequentially to promote cancer progression [2]. The uncontrolled cell division that represents cancer initiation, involves not only deregulated control of cell proliferation and resistance to apoptosis, but also changes in energy metabolism in order to fuel cell growth and division. For instance, normal cells under aerobic conditions process glucose, first to pyruvate via glycolysis in the cytosol and thereafter to carbon dioxide in the mitochondria; under anaerobic conditions, glycolysis is favoured. Tumour cells, even in the presence of oxygen limit their energy metabolism largely to a state of aerobic glycolysis, also known as the Warburg effect [2]. At the same time, to ensure nutrient and oxygen delivery, as well as to remove metabolic wastes and carbon dioxide from the constantly growing mass, it is fundamental to form new blood vessels through the angiogenesis process [2].

Another hallmark of cancer influenced by *HOX* genes, with a dual function in tumorigenesis, is inflammation [2,26]. While in normal cells, the inflammation process is carried out by innate immune cells to fight infections and heal wounds, in cancer cells it can instead inadvertently work to support tumour growth [2]. Cell movement, which drives the migration and invasion processes, is dependent on the supply of matrix-degrading enzymes such as metalloproteinases (group of enzymes that can break down proteins, such as collagen, that are normally found in the spaces between cells). Finally, the presence of cancer stem cells, with extensive growth and differentiation abilities, give phenotypic heterogeneity for tumour cells in order to enable the survival of specific cell subpopulations that are resistant to therapy and capable of regenerating the tumour once therapy has been halted [2].

An example of *HOX* gene alteration in cancer, relates with the increased expression of *HOXA9* in the most aggressive acute leukaemia and its potential as predictive of poor prognosis [28]. Moreover, it was demonstrated that *HOXA13* protein physically links to the translation initiation factor eIF4E, frequently overexpressed in cancer and described as a strong promoter of tumour growth and angiogenesis [29]. EIF4E acts on the exportation of specific oncogenes mRNAs from the nucleus to cytoplasm, such as c-Myc, FGF-2, ODC and CCND1. Thus, the deregulation of this *HOX* gene could facilitate the mRNA nuclear export of eIF4E-dependent oncogene transcripts, as observed in hepatocellular carcinomas (HCCs) [30]. *HOX* genes are also actively expressed in adipose tissue, which has emerged as an important supportive tissue for cancer proliferation and progression [26]. Half of the KRAS-mutant NSCLC (non-small-cell lung cancers) express the homeobox protein HOXC10, which triggered tumour regression in xenografts and PDX (patient-derived xenografts) models in vivo [31]. Upregulation of *HOXA10* expression plays a key role in colorectal cancer development and could be considered as a new biomarker that indicates poor prognosis [32]. *HOXC8* upregulation is inversely related to pancreatic ductal adenocarcinoma progression and metastasis formation and therefore could be explored as a marker for pancreatic cancer progression [33]. Moreover, some of the pathways involved in metastasis, from invasion to colonization, are affected by the HOX proteins through mechanisms that modulate their expression, such as *HMGA2*/*TET1*/*HOXA* and TGFβ, or by the interference of microRNAs and lncRNAs regulators [26]. However, further studies are required to consolidate their real potential as therapeutic targets [34].

In this context, several studies are now being performed to evaluate HOX proteins strength as therapeutic targets in cancer therapy. The synthetic peptide HXR9 (recently replaced by HTL001), that blocks the binding of HOX proteins to PBX cofactors and, consequently prevents the binding of these dimers to the DNA, is one of the approaches that are under evaluation [34,35]. Cancer cells’ sensitivity to this peptide is highly correlated to their *HOX* expression profile, albeit the subset of *HOX* genes which act as oncosuppressors is not affected [34]. This peptide has shown to be effective in oesophageal and oral squamous cell carcinomas [36,37], melanoma [35,38], ovarian cancer (OC) [39], breast cancer [40], meningioma [41], prostate cancer [42], and leukaemia [43]. The use of RNA interference mechanisms [44,45] and the control of *HOX* methylation status [46] are additional tools that can control *HOX* expression with therapeutic function. Therefore, the manipulation of *HOX* genes expression emerges as a promising strategy to prevent invasion and metastasis.

## 2. Invasion and Metastasis

Tumour formation and progression are multistep processes that depend on the accumulation of genetic and epigenetic alterations generating cells with great genetic instability and culminating with its uncontrolled proliferation escaping from apoptosis and immune system response [2,47]. Subsequently, the progressive increment of the tumour mass size associated with the activation of metalloproteinases (MMPs), which generates gaps in the barriers between tissues, allow cancer cells to invade and disseminate throughout adjacent normal tissues (invasion process), forming secondary tumours [2,48,49]. The development of metastasis is far more complex because, besides invading adjacent tissues, cancer cells travel through blood and lymphatic vessels, detect the premetastatic niche (PMN), and initiate the formation of new tumour masses in distant sites [6,47,50,51].

Therefore, the metastization process depends on the coordinated support of specific cellular programs such as angiogenesis, alterations of cell adhesion, tumour cell motility and secretion of proteolytic enzymes [52]. The dislocation of cancer cells throughout the blood vessels is achieved through the loss of their cell-to-cell adhesion molecules in a process known as epithelial-mesenchymal transition (EMT) [52]. Cancer cells change from epithelial to mesenchymal phenotype enabling cell motility due cytoskeleton modifications that allow cell movement from primary tumour mass to the blood and lymph vessels (intravasation) [53]. When in circulation, metastatic tumour cells adhere to the basement membrane of a new tissue site (extravasation) by mechanisms not completely understood. This process involves the action of glycoproteins such as fibronectin, type IV collagen and laminin. Subsequently, metastatic tumour cells secrete degradative enzymes, or induce host tissues to produce them, to degrade the ECM of the new tissue site [54]. Finally, cancer cells are driven across the basement membrane and stroma within the region where proteolysis occur, starting its division and establishing tumour masses in this new location with or without the reversion of the mesenchymal phenotype through mesenchymal to epithelial transition (MET) process [1] (Figure 1).

## 3. Five Mechanisms of HOX Deregulation Affecting Invasion and Metastasis of Cancer Cells

### 3.1. *HMGA2/TET1/HOXA* Signalling Pathway

The ten–eleven translocation family (TET) of methylcytosine dioxygenases can induce DNA demethylation and is associated with tumorigenesis in many cancer types, given that its action alters the regulation of transcription and affects the expression of many genes [55,56,57]. These enzymes catalyse the hydroxylation of 5-methylcytosine (5mC) found in methylated DNA sequence, which results in a 5-hydroxymethylcytosine (5hmC). Then, TET can further catalyse oxidation of 5hmC to 5-formylcytosine (5fC) that is also oxidized forming 5-carboxycytosine (5caC). Finally, by the action of a terminal deoxynucleotidyl transferase (TdT), the 5caC is converted to an unmodified cytosine [58] (Figure 2).

An association was also found between TET enzymes and *HOX* gene expression. For example, in breast cancer, TET1 can demethylate its own promoter and the promoter of *HOXA* genes, enhancing its own expression and stimulating *HOXA7* and *HOXA9* expression (Figure 2) [56]. *HOXA9* overexpression has been correlated to a less invasive behaviour of breast cancer cells [56,59]. The TETs are also associated with the nonhistone architectural transcription factor “high mobility group AT-hook 2” (*HMGA2*) and its inhibition can induce *TET1* expression [56]. *HMGA2* can modulate the transcription of several genes binding to AT-rich sequences and altering the chromatin structure and consequently affecting a variety of processes, including the cellular cycle, DNA damage repair, apoptosis, senescence, EMT and telomere restoration [58]. Therefore, the effect of *HMGA2 in TET1* expression and the subsequent effect of TET1 in *HOXA9* expression is known as the *HMGA2/TET1/HOXA9* signalling pathway, which has a role in the invasion and metastasis of cancer cells (Figure 2). In breast cancer, when HMGA2 is depleted, an induction of the *TET1* and *HOXA9* genes occurs inhibiting invasion and metastization [56]. The opposite effect could also occur, given that *HMGA2* overexpression reduces *TET1* and *HOXA9* expression, supporting invasion and metastization [60]. These data highlight tissue-specificity of *HOX* gene functions, not only in embryonic development but also in cancer [15,17].

### 3.2. TGFβ Signalling Pathway

*HOX* proteins interfere with invasion and metastasis of cancer cells due to their ability to affect the expression of *TGFβ* genes, which are multifunctional cytokines secreted by different cells, with important roles in embryonic development controlling cell behaviour, namely cell proliferation, differentiation, morphogenesis, tissue homeostasis and regeneration [61]. The direct binding of HOX proteins to *TGFβ* genes promoters, activating their expression, was identified for *HOXB7* and *HOXC8* [62,63]. This results in the coding of TGFβ ligands, proteins that will be secreted to the ECM [64] (Figure 3). Thus, the dysfunction of TGFβ seems to be also associated with cancer progression, as a consequence of *HOX* genes expression deregulation [65]. *HOXB7* overexpression, for example, induces invasive and metastatic breast cancer by activating the TGFβ signalling pathway [62]. In HCCs, downregulation of *HOXB5* inhibits TGFβ, which induces migration and invasion [66]. In NSCLC, *HOXC8* promotes proliferation and migration through transcriptional upregulation of TGFβ1 [63]. When in the ECM, TGFβ ligands bind to TGFβ receptor type II in the cell surface and a cascade of processes is initiated (Figure 3), causing modifications in the expression of genes related with cell proliferation, apoptosis, metastasis and angiogenesis [67].

### 3.3. HOX Proteins Effect in the Epithelial to Mesenchymal Transition (EMT)

EMT is a fundamental cellular process for non-disease conditions, such as embryogenesis and wound healing; however, it can also favour malignant processes, such as cancer progression [68]. Intercellular structures and cell–cell adhesion molecules are key factors in maintaining a well cohesive normal tissue or even a coherent primary tumour mass. Abnormalities in these structures can lead to cell detachment from the primary tumour and enhance the potential for dissemination and metastatic spread of cancer cells to secondary locations [69].

During EMT, epithelial cells are transcriptionally reprogrammed, which results in a decrease of cell–cell, cell–basement membrane adhesion and an increase of cell migration or invasion [70]. The epithelial cells usually present apical–basal polarity and are maintained together laterally by tight and adherens junctions, formed by epithelial cadherin, CDH1, and desmosomes. Cell junctions with basement membranes are maintained by hemidesmosomes [53] (Figure 4). To maintain these junctions and cell polarity, cells express epithelial adhesion molecules, such as occludin, claudins, α6β4 integrin and cytokeratins. Therefore, for EMT to occur, two parallel processes must take place: (a) repression in the expression of epithelial molecules by EMT-inducing transcription factors expression, namely ZEB, SNAIL and TWIST, and (b) activation of mesenchymal molecules expression, as N-cadherin, vimentin, fibronectin, β1 and β3 integrins and MMPs [53,71]. As a consequence of the EMT process, cells acquire motility and invasive capacities [72] (Figure 4). Thus, EMT has significant importance in epithelial cancers, since it confers a higher tumour-progression potential, as invasion and metastatic capabilities, to cancer cells [53]. This new mesenchymal cell status can be reversible, and cells can recover their epithelial characteristics in a process called mesenchymal-to-epithelial transition (MET) [72].

Currently, there are some reports proposing an involvement of HOX proteins in the EMT process, with an impact on the capacity of *invasion and metastasis of cancer cells* [73,74,75,76]. It has been demonstrated that *CDH1* expression is transcriptionally upregulated by the direct binding of HOXA5 to their promoter sequence and that the knockdown of *HOXA5* in mammary cells leads to a loss of epithelial traits, increased stemness and cell plasticity, and acquisition of more aggressive phenotypes [77]. *HOXC6* was shown to promote invasion of HCCs by driving EMT through the negative regulation of CDH1 expression and being positively associated with vimentin and MMP-9 expression [78,79]. Still in HCC, HOXD9 was shown to interact with the promoter region of *ZEB1/ZEB2* increasing its expression. ZEB1 knockdown, in turn, inhibited HOXD9-induced migration and invasion, as well as EMT [78].

*HOXA10,* which is reported as an EMT inhibitor, when downregulated in endometrial carcinomas favoured their invasive behaviour by reducing the expression of CDH1 [80]. In gastric carcinomas (GC), the downregulation of *HOXB9* has been associated with malignancy and metastasis while its re-expression was suggested to cause inhibition of cell proliferation, migration, invasion and induction of MET [73]. In cervical cancers, *HOXA10* and *HOXB13* were shown to influence the EMT. *HOXA10* expression was positively associated with CDH1 and negatively associated with vimentin expression, inhibiting EMT. For HOXB13 the opposite trend was observed with a negative effect on CDH1 and a positive effect on vimentin expression [75]. Additionally, it has been demonstrated that *HOXB13* promotes lung adenocarcinoma cell growth, local invasion, and metastasis to the liver in nude mice regulating *ABCG1*, *EZH2* and *Slug* [78,81]. In breast cancer, *HOXB7* overexpression can promote tumour migration and invasion through the induction of EMT in epithelial cells, by reducing the expression of epithelial proteins, as Claudin-1 and Claudin-7, together with mislocalisation of Claudin-4 and CDH1 and increased expression of mesenchymal proteins: vimentin and alpha-smooth muscle actin (α-SMA) [82] (Figure 4). A detailed list of the proposed roles of the HOX proteins in EMT is presented in Table 1.

### 3.4. MicroRNA’s Interference

MicroRNAs (miRs) are an abundant class of noncoding RNAs (with 21 to 25 nucleotides), which have a crucial role in post-transcriptional gene regulation, including *HOX* genes expression (Table 2) [98]. This regulation of the *HOX* genes by miRs is favoured by the location of miRs coding regions in the *HOX* gene clusters [99]. Thus, the role of the *HOX* genes in invasion and metastasis of cancer cells might be also dependent on their regulatory miRs. In pancreatic cancer, for example, it has been described that the downregulation of *HOXA1*, *HOXB1* and *HOXB3* by miR-10a is associated with the formation of metastasis [100,101]. Furthermore, miR-100 can directly target the 3′-UTR of *HOXA1*, which reduces the chemotherapy response of small-cell lung cancer [102]. Another example of miR interference is the downregulation of *HOXD10* by miR-10b, which favours the invasive and metastatic behaviour of breast cancer cells [103]. Besides, downregulation of *HOXD10* by miR-10b also causes an increase in prometastatic gene products, such as MMP-14 and RHOC, contributing to the acquisition of metastatic phenotypes in epithelial OC cells [104]. Moreover, miR-7 and miR-218 epigenetically control tumour suppressor genes RASSF1A and Claudin-6 by downregulating HOXB3 in breast cancer [105]. In HCC, miR-203a negatively targets *HOXD3* resulting in an inhibition of cell invasion, metastasis and angiogenesis [96]. On the other hand, miR-224 directly targets and downregulates *HOXD10*, boosting tumour progression and invasion in HCC [106]. In addition, *HOXB7* downregulation promoted by mR-196b-5p leads to metastization in colorectal cancer [107]. Some studies have also connected miR interference with *HOXA5* downregulation, considered as a tumour suppressor gene in several cancer contexts [15]. For instance, miR-181d-5p targets CDX2, a transcription factor binding to HOXA5 promoter, consequently leading to EMT in MCF-7 breast cancer cells [108]. Moreover, in NSCLC, miR-196a antagonizes the inhibitory tumour growth effect of HOXA5, partially contributing to an increasingly invasive phenotype [109]. Another noteworthy miR target is HOXC10, shown to be involved in the EMT in both GC and ovarian OC [94,95]. Recent studies have shown that miR-136 and miR-222-3p act as an inhibitor of *HOXC10*, which impairs the EMT and consequently leads to less perineum and hepatic metastasis in GC and OC, respectively [94,110].

The relationship between *HOX* gene expression and miRs regulation is far more complex than initially described. In fact, increasing evidence suggest that HOX proteins can themselves regulate the expression of miRs, which has an impact on invasion and metastasis abilities of cancer cells. In breast cancer and gliomas, *HOXD10* downregulation results in the downregulation of miR-7 and upregulation of p21-activated kinase 1 expression (PAK1), which culminates in higher aggressiveness of these cancer types facilitating invasion and metastasis [111].

### 3.5. lncRNA’s Interference

Over the last decade, several long noncoding RNAs (lncRNAs) were proposed to play essential roles in gene regulation, both at pre- and post-translational levels, and described to contribute to tumour progression and metastasis [112]. A total of 231 lncRNAs have been identified as products of the HOX clusters, amongst which *HOXD-AS1*, *HOXA-AS2*, *HOTAIRM1*, *HOTTIP*, and *HOTAIR*, were reported to be involved in cancer regulation [113].

The HOX antisense intergenic RNA (HOTAIR) is an lncRNA that correlates with metastasis and poor prognosis in breast, colon and lung cancer [114,115,116]. This lncRNA is transcribed from the antisense strand of the *HOXC* cluster, repressing the transcription of the *HOXD* cluster [45]. The induction of HOTAIR expression in breast and lung cancer cell lines promotes growth in soft agar and invasion in Matrigel [45]. For example, in breast cancer, HOTAIR downregulates *HOXD10* favouring the invasive and metastatic behaviour (Table 2) [103]. In gliomas, HOTAIR was described to promote invasion through upregulation of *MMP-7*, *MMP-9* and *VEGF* [117]. In addition, HOTAIR has been identified as a key modulator in chromatin dynamics, partaking simultaneously in histone methylation and deacetylation, but also on the antagonistic process [118,119]. In fact, the knockdown of HOTAIR in GC significantly reverses EMT by increasing the expression of *CDH1* and inducing the loss of the PRC2 complex activity (consisting of H3K27 methyltransferase EZH2, SUZ12 and EED) [120]. In cervical cancer, the downregulation of HOTAIR results in an increased expression of *CDH1* and a decreased expression of *CTNNB1*, *vimentin*, *Snail* and *Twist*. In contrast, *HOTAIR* overexpression promotes VEGF and MMP-9 protein expression [121]. In osteosarcoma, *HOTAIR* suppression significantly reduces the migration and invasion, decreases the expression *of MMP-2* and *MMP-9* and increases the expression of *CDH1*. This occurs through regulation of RAC-α serine/threonine protein kinase target in rapamycin signalling pathway [122], whereas in gliomas, this lncRNA increases *MMP-7* and *MMP-9* expression by upregulating the Wnt/β-catenin pathway [123]. Moreover, in renal cell carcinomas [124] and HCC [125], upregulation of HOTAIR leads to increased cell invasiveness and metastatic capacity. Therefore, it would be fair to infer the involvement of this lncRNA on the epithelial-mesenchymal transition.

A recent and increasingly number of studies have suggested that HOTAIR can interact with miRNAs to regulate gene expression and cancer progression. In fact, HOTAIR may sponge miRs, by competing for its target sites, thus inhibiting its effects. In breast cancer, HOTAIR was shown to promote invasion and metastasis by sponging miR-601, which consequently downregulated its target, ZEB1 [126]. In OC, *HOTAIR* actively competes with miR-260 upregulating CCND1 and CCND2 dynamically sprouting OC progression [127]. Moreover, HOTAIR can act as a molecular sponge of miR-204, leading to an increase in HOXC8, promoting proliferation, migration and invasion in oesophageal cancer cells [128], whereas in cholangiocarcinoma, this process occurs through the modulation of miR-204-5p’s interaction with HMGB1 [129]. Notwithstanding, *HOTAIR* promotes GC cell growth and metastasis by interfering in the miR-217/GPC5 [130], miR-1277-5p/COL5A1 [131], and miR-618/KLF12 axis, being that the last one induces metastasis by promoting the PI3K/ATK signalling pathway [132]. Recent findings have also related HOTAIR interference to Trastuzumab resistance in GC, through the sponging of miR-3030 and consequent upregulation of ERBB4 [133], which is responsible for activating the PI3K/ATK signalling pathway [134]. Several HOTAIR-dependent interferences in miR pathways have also been described in glioma [123], HCC [135,136,137] prostate cancer [138,139] and squamous cell carcinomas [140,141,142,143].

Despite HOTAIR’s extensively described role in carcinogenesis, other lncRNAs transcribed from the HOX clusters should not be overlooked in the cancer context. HOXA cluster antisense RNA 2 (*HOXA-AS2*), transcribed between the *HOXA3* and *HOXA4* genes, has been implicated in oncogenic promotion in pancreatic cancer, breast cancer [144,145], colorectal cancer [146], GC [147,148], bladder cancer [149], HCC [150], thyroid cancer [151], gallbladder carcinoma [152], lung cancer [153,154,155], osteosarcoma [156], glioma [157], and leukaemia [158,159]. Another lncRNA transcribed between *HOXA1* and *HOXA2*, the HOXA transcript antisense RNA myeloid-specific 1 (*HOTAIRM1*), appears upregulated in endometrial and pancreatic cancers, NSCLC, glioma and glioblastoma, but downregulated in HCC, colorectal, OC, GC and papillary thyroid cancers, as well as head and neck tumours [160].

Another lncRNA transcribed from the genomic region of *HOXA* locus is HOXA distal transcript antisense RNA (*HOTTIP*) that directly binds the *HOXA* locus leading to H3K4 methylation and activating the transcription of *HOX* genes. HOTTIP has been described to exert oncogenic roles in different malignancies such as head and neck tumours, oesophageal carcinomas, colorectal, lung and breast cancers (Table 2) [161]. Furthermore, HOXD cluster antisense RNA 1 (*HOXD-AS1*) also named HAGRL, transcribed on the antisense direction between *HOXD1* and *HOXD3*, appears to be deregulated in many cancers. In fact, this lncRNA is overexpressed in several malignancies, promoting processes such as cell proliferation, metastasis, and drug resistance [162], acting as a molecular sponge for different miRs [163] or by interfering in chromatin modulation [164]. Amongst these malignancies are cervical cancer [165,166], GC [167], colorectal cancer [168], breast [169], OC [170], and HCC [171,172]. HOXD-AS1 also promotes cell migration and invasion in NSCLC by sponging miR-133b and promoting MMP-9 expression [173], albeit it plays an opposite stand in lung adenocarcinoma by recruiting DNMT1 to the promoter of E2F1, silencing it, and consequently suppressing cell proliferation [164].

**Table 2 cancers-13-00010-t002:** Noncoding RNAs that regulate HOX genes and/or are transcribed from HOX loci with impact on breast cancer progression.

Non-Coding RNA	HOX Gene Affected	Effect On Breast Cancer Cells	Reference
miR-10b	↓ HOXD10	↑ Aggressiveness	[103]
miR-7/-218	↓ HOXB3	⊗ Cell cycle/clone formation	[105]
miR-181d-5p	↓ HOXA5	↑ Aggressiveness, promotes EMT	[108]
HOTAIR	↓ HOXD10	↑ Invasion/ metastasis	[103]
HOXA-AS2 *	NA	↑ Oncogenic behaviour	[144]
HOTTIP	↑ HOXA	↑ oncogenic behaviour	[161]
HOXD-AS1 (HAGRL) *	NA	↑ Oncogenic behaviour	[169]

⊗, Inhibition; ↑, Increase; ↓, Decrease; *, Noncoding RNAs transcribed from HOX genes loci; NA, Not Applicable.

## 4. Conclusions

Invasion and metastasis are the leading causes of cancer-related death, as they reflect the last stages of cancer. Therefore, the development of targeted therapies to prevent and/or lagging invasion and metastasis are crucial to improve the clinical outcomes of cancer patients. *HOX* genes are in the spotlight of cancer research due to their effects on key molecules related to angiogenesis (VEGFR), invasion (MMPs), cell adhesion (CDH1, vimentins, claudins), and transcription factors involved in the EMT (Twist, Snail, Zeb). These genes were also implicated in the control of mRNAs nuclear export achieved by their impact on eIF4E. In addition, HOX genes were shown to be active in the adipose tissue, an important supportive tissue for tumour progression. However, the identification of HOX targets and their function in different types of cancer is incomplete. Furthermore, microRNAs and lncRNAs transcribed from the HOX *loci* can also be part of the complex molecular processes by which HOX genes affect invasion and metastasis in cancer. Over the last years, the use of the HXR9 peptide has yielded promising results in different types of tumours and could be an innovative tool in different stages of cancer progression.

## 5. Future Perspectives

Different aspects of the *HOX* genes function in cancer have been studied and diverse approaches have already been tested to allow the use of the *HOX* genes as therapeutic targets for cancer treatment. Here, we propose three take-home messages: (1) HOX have crucial roles in two of the most aggressive processes in cancer development, invasion and metastization, responsible for the majority of cancer deaths; (2) *HOX* genes can affect invasion and metastization by diverse mechanisms, and possibly more are yet to be discovered; (3) the control of *HOX* genes expression and/or its targets may represent a promising strategy to prevent invasion and metastasis and decrease the number of cancer-related deaths.

## Figures and Tables

**Figure 1 cancers-13-00010-f001:**
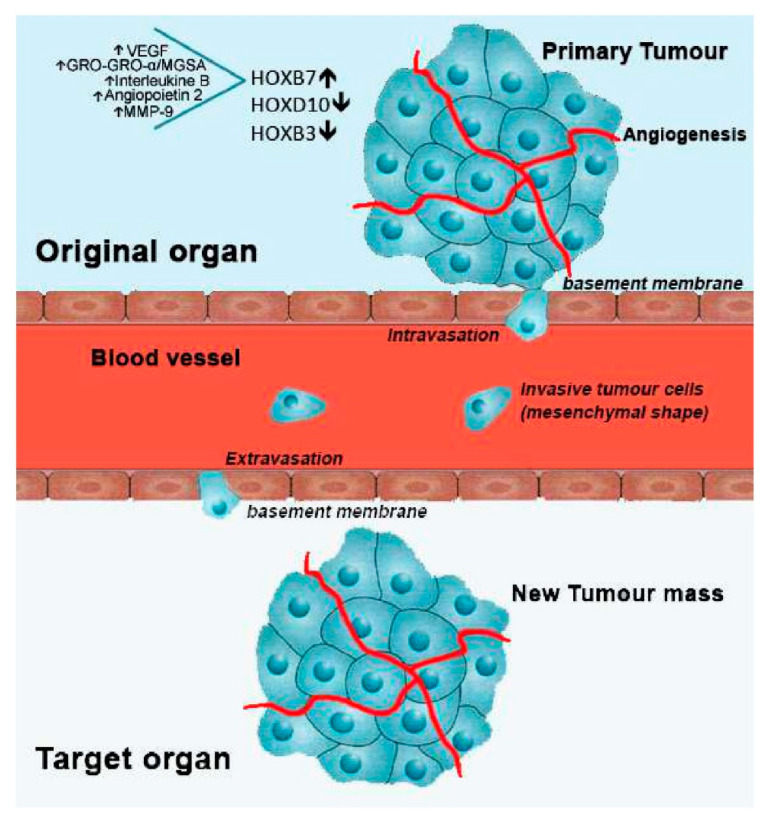
Metastatic process and involved *HOX* genes, regarding breast cancer. After cell transformation, a mass is rapidly formed in a specific organ of the body, designated as primary tumour. With the formation of new blood vessels surrounding the primary tumour (angiogenesis), growth is stimulated by the flux of nutrients and oxygen. Then, cells from the primary tumour may acquire motility going through the basement membrane into the blood vessels (intravasation). Once in circulation, metastatic tumour cells stop its progress adhering to the basement membrane of a new tissue site (extravasation). Finally, cancer cells start to proliferate in the new location, forming a new tumour mass. In breast cancer, overexpression of *HOXB7*, along with the upregulation of the indicated molecules, and downregulation of *HOXD10* and *HOXB3* have been linked to the metastatic process.

**Figure 2 cancers-13-00010-f002:**
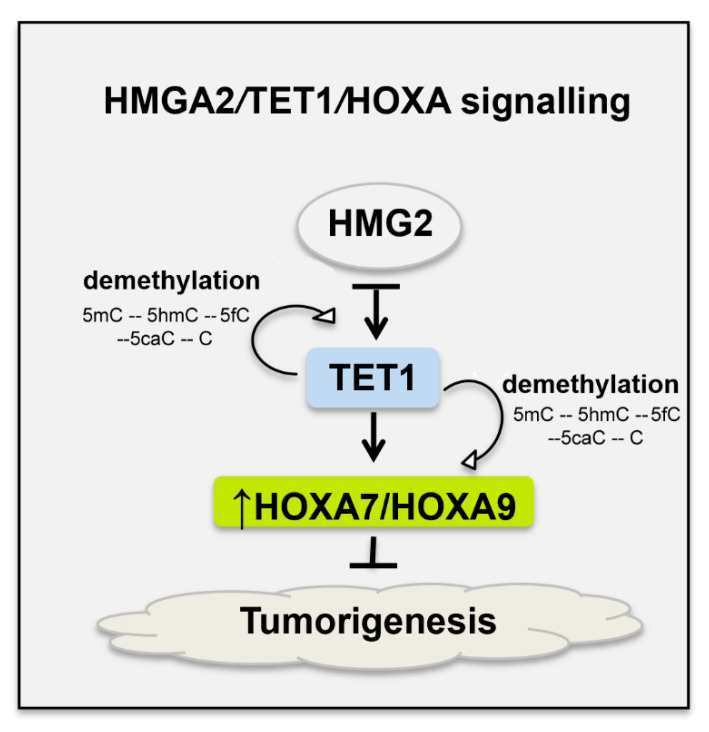
HMGA2/TET1/HOXA signalling in breast cancer cells. Depletion of the architectural transcription factor high mobility group AT-hook 2 (HMGA2) induces TET1 (ten–eleven translocation methylcytosine dioxygenases). This molecule initiates demethylation of DNA and is associated with tumorigenesis in many cancers. TET1 binds and demethylates its own promoter and the promoters of *HOXA7* and *HOXA9*, increasing their expression, which may result in the inhibition of breast cancer growth and metastasis, already demonstrated for HOXA9 using mouse xenografts [35]. The demethylation promoted by TET1 allows the conversion of 5-methylcytosine (5mC) to 5-hydroxymethylcytosine (5hmC) and this to 5-formylcytosine (5fC), which in turn is converted in to 5-carboxycytosine (5caC) and at the end in an unmodified cytosine (C).

**Figure 3 cancers-13-00010-f003:**
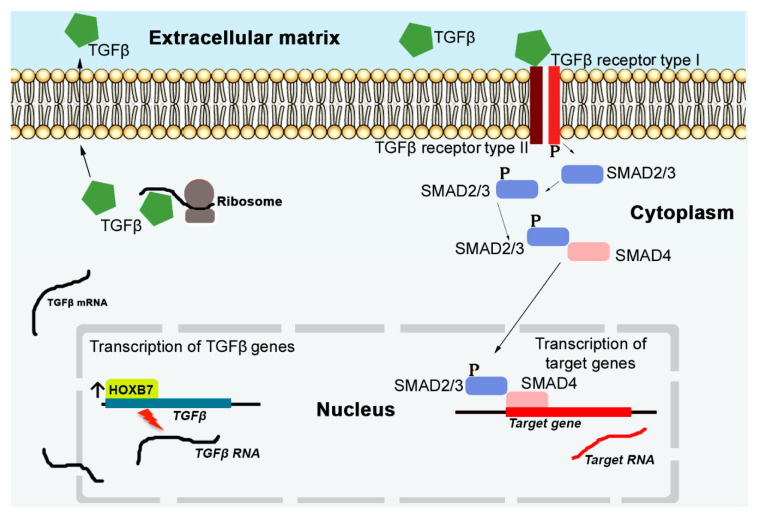
TGFβ signalling pathway in breast cancer cells. The reception of the TGFβ signals by the TGFβ receptor type II allows the phosphorylation of the TGFβ receptor type I. Subsequently this favour the phosphorylation of SMAD2/3 proteins that can in turn interact with the protein SMAD4. This complex of proteins controls the expression of TGFβ target genes, affecting cellular processes, such as proliferation, apoptosis, metastasis, and angiogenesis. The expression of the TGFβ’s genes is regulated by HOX transcriptional factors, as HOXB7 in breast cancer. After its transcription (nucleus) and translation (cytoplasm), TGFβ ligands are exported to the ECM, where they recognize and are recognized by TGFβ receptors type II.

**Figure 4 cancers-13-00010-f004:**
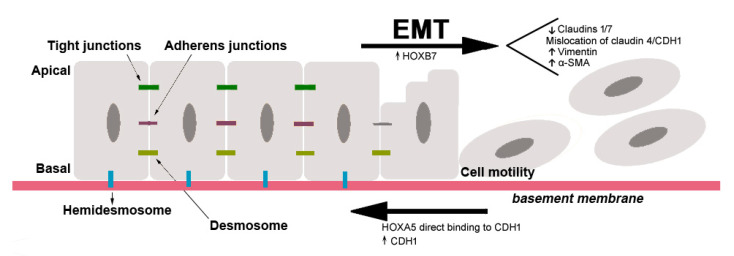
Examples of *HOX* genes involved in the activation or inhibition of the epithelial to mesenchymal transition (EMT) in breast cancer. Epithelial cells present apical–basal polarity and are maintained together by tight junctions, adherens junctions and desmosomes. Hemidesmosomes maintain epithelial cells together with basement membrane. At the end of this process, epithelial cells present mesenchymal characteristics as the absence of apical–basal polarity and presence of motility capacity. HOXA5 is known to inhibit the EMT via upregulation of CDH1 in the adherens junctions, while HOXB7 induce the EMT downregulating the epithelial proteins: Claudin-1, Claudin-7, Claudin-4 and CDH1, and upregulating vimentin and α-SMA.

**Table 1 cancers-13-00010-t001:** Proposed role of the HOX proteins in the EMT and MET in distinct cancer types.

HOX Protein	Type of Cancer	Effect	Reference
HOXA1	Prostate cancer	Induction of EMT via downregulation of *CDH1* and upregulation of *MMP-3* and *Snail*.	[83]
HOXA5	Breast cancer	Inhibition of EMT via upregulation of *CDH1*.	[77]
HOXA9	Breast Cancer	Less invasive phenotype inclusive in claudin-low cells.	[56,59]
HOXA10	Endometrial carcinoma	Inhibition of EMT via upregulation of *CDH1*.	[80]
	Cervical cancer	Inhibition of EMT via upregulation of *CDH1* and downregulation of *vimentin*.	[75]
	Oral squamous cell carcinoma	Inhibition of EMT via upregulation of *CDH1* and downregulation of *N-cadherin*.	[84]
	Ovarian cancer	Inhibition of EMT via downregulation of *vimentin*, and *MMP-9*.	[85]
	Glioma	Induction of EMT via activation of Wnt/beta-catenin/TGF-β pathways.	[86]
HOXA13	Oesophageal squamous cell carcinoma	Induction of EMT via downregulation of *CDH1* and upregulation of *Snail*.	[87]
	Gastric cancer	Induction of EMT via downregulation of *CDH1* and upregulation of *N-cadherin* and *vimentin*.	[88]
HOXB5	Breast cancer	Induction of EMT via downregulation of *CDH1* and upregulation of *vimentin* and *Snail2*.	[89]
	Non-small-cell lung cancer	Induction of EMT via activation of Wnt/beta-catenin pathway.	[90]
HOXB7	Breast cancer	Induction of EMT via downregulation of Claudin-1 and Claudin-7, mislocalisation of Claudin-4 and CDH1, and upregulation of *vimentin* and *α-SMA*.	[82]
HOXB8	Colorectal carcinoma	Induction of EMT via downregulation of *CDH1* and upregulation of *vimentin*, *N-cadherin*, *Twist*, *Zeb1* and *Zeb2*.	[91]
	Ovarian cancer	Inhibition of EMT via downregulation of *vimentin*, and *MMP-9*.	[85]
	Gastric carcinoma	Induction of MET.	[73]
HOXB9	Lung adenocarcinoma	Induction of EMT via downregulation of *CDH1* and upregulation of *N-cadherin* and *vimentin*.	[92]
	Breast cancer	Induction of EMT via activation of the TGF-β pathway.	[93]
HOXB13	Cervical cancer	Induction of EMT via down regulation of *CDH1* and upregulation of *vimentin*.	[75]
	Lung adenocarcinoma	Induction of EMT via ABCG1, EZH2 and *Slug* regulation.	[78,81]
HOXC6	Hepatocellular carcinoma	Induction of EMT via positive regulation of *CDH1* and negative regulation of *vimentin* and *MMP*-9.	[78,79]
HOXC10	Ovarian cancer	Induction of EMT via *Slug* regulation.	[94]
	Gastric cancer	Induction of EMT via activation of the MAPK pathway.	[95]
HOXD3	Hepatocellular carcinoma	HOXD3 can directly target the promoter region of *VEGFR* and increase its expression.	[96]
HOXD9	Hepatocellular carcinoma	Induction of EMT via upregulation of *ZEB1* and *ZEB2* interaction	[74,78]
	Colorectal carcinoma	Induction of EMT via upregulation of *TWIST* and *SNAIL*.	[76]
HOXD10	Oesophageal squamous cell carcinoma	Inhibition of EMT via upregulation of *CDH1* and downregulation of *N-cadherin* and *vimentin*.	[97]

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
