# Peer review of "Roles of the HOX Proteins in Cancer Invasion and Metastasis"

_cancers, 2020, doi:10.3390/cancers13010010_

Round 1

Reviewer 1 Report

The review by Paço et al provides an overview of the role of HOX proteins in cancer invasion and metastasis. While I appreciate the efforts of the authors  in writing this review, I have some major and minor comments:

Major comments:

  1. In line 49, the authors refer to Epigenetic Modification but do not provide any explanation for it. A short paragraph is required to explain epigenetic mechanisms.
  2. The sentence from line 55-58 introduces a lot of concepts/mechanisms without proper introduction. A few sentences are required to explain uncommon words such as autophagy, apoptosis, the type of cellular metabolism of differentiation that they refer to.
  3.  Line 65, talks about eIF4E. There is no explanation of the protein translation mechanism and the new proteins keep being introduced into the review without proper explanation. A short section should be included to describe role of protein translation with regards to the topic of the review.
  4. While the authors talk in one sentence about PBX (line 81), there are no sections on HOX co-factors. Which PBX are the authors referring to? A subheading about the HOX o-factor proteins should be included. Many of these co-factors are also involved in cancer (i.e. PBX, MEIS, PREP family of proteins that make up the TALE proteins).
  5. Line 80 talks about specific paraloug groups without discussing what is a HOX cluster, what are the 13 parlour groups, how they are expressed in a collinear fashion during development. This seems to be out of context. A subheading should be included to discuss these matters: HOX clusters, HOX paraloug groups, their redundancy in function, role in cancer, and collinear expression pattern in development with a unidirectional chromatin opening.
  6. Line 131-132 the section on TET proteins is not fully accurate. These proteins are responsible for active DNA demethylation through several steps. This has to be described accurately, from 5-mC to 5-hmC, and further oxidization, and proper process to remove the methyl group. The authors should properly explain this concept in the review and provide a proper step-by-step schematic in the Figure 2. This has been over-simplified.
  7. A better interpretation of the singling molecules, as they appear out of context. A better presentation of the importance of the discussed molecules and pathways, with a short but sufficient introduction at the beginning of the respected sections is required.

Minor comments:

  1. There are formatting and spelling errors throughout the manuscript that should be corrected.
  2. The authors should choose only one spelling format and keep it consistent for the word: "tumour" versus "tumor" that is used in both styles. Examples on lines 26 and 30, ...

Author Response

Reviewers’ comments:

REVIEWER 1

The review by Paço et al provides an overview of the role of HOX proteins in cancer invasion and metastasis. While I appreciate the efforts of the authors in writing this review, I have some major and minor comments:

Major comments:

  1. In line 49, the authors refer to Epigenetic Modification but do not provide any explanation for it. A short paragraph is required to explain epigenetic mechanisms.

A: We introduced an explanation about epigenetic mechanisms and the difference in relation to genetic alterations in the main text.

  1. The sentence from line 55-58 introduces a lot of concepts/mechanisms without proper introduction. A few sentences are required to explain uncommon words such as autophagy, apoptosis, the type of cellular metabolism of differentiation that they refer to.

A: We added an explanation about cell autophagy, proliferation, apoptosis, migration, invasion, metastization, differentiation, angiogenesis, metabolism, metalloproteinases regulation and inflammation.

  1. Line 65, talks about eIF4E. There is no explanation of the protein translation mechanism and the new proteins keep being introduced into the review without proper explanation. A short section should be included to describe role of protein translation with regards to the topic of the review.

A: We included in the main text information about the translation initiation factor eIF4E and how it impacts the exportation of specific oncogene mRNAs, from nucleus to cytoplasm. Throughout the main text we also explained the role and impact of all mentioned proteins in cancer, following the recommendation of the reviewer.

  1. While the authors talk in one sentence about PBX (line 81), there are no sections on HOX co-factors. Which PBX are the authors referring to? A subheading about the HOX o-factor proteins should be included. Many of these co-factors are also involved in cancer (i.e. PBX, MEIS, PREP family of proteins that make up the TALE proteins).

A: The introduction section was restructured in order to cover information about HOX co-factors, and the approaches already performed to use HOX as therapeutic targets as well as the comments pointed out in items 2 and 3.

  1. Line 80 talks about specific paralog groups without discussing what is a HOX cluster, what are the 13 paralog groups, how they are expressed in a collinear fashion during development. This seems to be out of context. A subheading should be included to discuss these matters: HOX clusters, HOX paralog groups, their redundancy in function, role in cancer, and collinear expression pattern in development with a unidirectional chromatin opening.

A: We added information about HOX gene genomic organization and transcription with the inclusion of a subheading in the introduction section.

  1. Line 131-132 the section on TET proteins is not fully accurate. These proteins are responsible for active DNA demethylation through several steps. This has to be described accurately, from 5-mC to 5-hmC, and further oxidization, and proper process to remove the methyl group. The authors should properly explain this concept in the review and provide a proper step-by-step schematic in the Figure 2. This has been over-simplified.

A: As suggested, we explained how TET enzymes act including all the steps of its action. We also include this information on Figure 2.

  1. A better interpretation of the singling molecules, as they appear out of context. A better presentation of the importance of the discussed molecules and pathways, with a short but sufficient introduction at the beginning of the respected sections is required.

A: As suggested, we performed a more complete presentation of the discussed molecules and pathways, with a short introduction at the beginning of each section.

Minor comments:

  1. There are formatting and spelling errors throughout the manuscript that should be corrected.

A: We corrected all the spelling mistakes in the manuscript, writing all the words in English from UK.

  1. The authors should choose only one spelling format and keep it consistent for the word: "tumour" versus "tumor" that is used in both styles. Examples on lines 26 and 30.

A: We corrected all the spelling mistakes in the manuscript, writing all the words in English from UK.

Reviewer 2 Report

The review by Paco and coauthors deals with an extensive field in cancer research comprising HOX genes and metastasis. They have included a lot literature and wrote an overall interesting and informative paper. However, I have some major and minor points which require basic changes in the manuscript.

major:

The structure of the manuscript is ordered in Introduction, Invasion and metastasis, Mechansisms, Conclusions. At the end of the introduction they announce the description of 5 mechanisms by which HOX genes impact invasion/metastasis. But it follows an aditional introduction. That is confusing. Please fuse all introducing parts into one paragraph. 

The topic of the review is the role of HOX genes in invasion and metastasis as mentioned in the title and in the abstract. Accordingly, 5 mechanisms were announced how HOX genes impact invasion/metastasis. However, the first paragraph deals with the deregulation of HOX genes by HMG/TET signalling and the fourth paragraf with their deregulation by micro RNAs. Therefore, these parts do not describe the announced mechanism. So, please skip these parts or reorganize the manuscript according to an altered topic of the review. 

The last paragraph deals with lncRNAs. However, the listed effects of the indicated lncRNAs via their HOX targets have mostly no impact in invasion/metastasis. Please focus on your main topic. 

minor:

Citation numbers 23 and 28 are identical.

Author Response

REVIEWER 2

The review by Paco and co-authors deals with an extensive field in cancer research comprising HOX genes and metastasis. They have included a lot literature and wrote an overall interesting and informative paper. However, I have some major and minor points, which require basic changes in the manuscript.

Major comments:

  1. The structure of the manuscript is ordered in Introduction, Invasion and metastasis, Mechanisms, Conclusions. At the end of the introduction they announce the description of 5 mechanisms by which HOX genes impact invasion/metastasis. But it follows an additional introduction. That is confusing. Please fuse all introducing parts into one paragraph.

A: We restructured the manuscript organization. We now start with an introduction about the topic (section 1), with subheadings 1.1 (HOX genes genomic organization and transcription) and 1.2 (Hox genes and cancer). After that, we describe the invasion and metastasis processes (section 2), and the five mechanisms by which HOX genes affect cancer cells invasion and metastasis (section 3). At the end, we present a section of conclusions and future perspectives (sections 4 and 5).

  1. The topic of the review is the role of HOX genes in invasion and metastasis as mentioned in the title and in the abstract. Accordingly, 5 mechanisms were announced how HOX genes impact invasion/metastasis. However, the first paragraph deals with the deregulation of HOX genes by HMG/TET signalling and the fourth paragraph with their deregulation by micro RNAs. Therefore, these parts do not describe the announced mechanism. So, please skip these parts or reorganize the manuscript according to an altered topic of the review.

A: As suggested, we restructured the manuscript organization as above described.

  1. The last paragraph deals with lncRNAs. However, the listed effects of the indicated lncRNAs via their HOX targets have mostly no impact in invasion/metastasis. Please focus on your main topic.

A: We kept the subsection on lncRNAs since they can affect HOX gene expression and, through them, impact invasion and metastasis formation. Several examples are presented in table 2 in this subsection.

Minor Comments:

  1. Citation numbers 23 and 28 are identical.

A: We corrected citations 23 and 28.

Round 2

Reviewer 1 Report

In reviewing the revised version of this manuscript, I did not find the sections that were added on my previously required major points 1 and 2 (any new section on epigenetic mechanisms, authophagy, apoptosis, etc). There are limited changes that I found to be included in the revised V2 manuscript, and the changes were not marked. I suggest a highlighted text, different color of the text, or track change for all revised sections.

A careful spelling and formatting check is required:

i.e. Section 1.2 should read HOX genes and cancer not HOX gene and cancer. There are 39 HOX genes in mammals (not one HOX gene).

Author Response

REVIEWER 1

  1. “In reviewing the revised version of this manuscript, I did not find the sections that were added on my previously required major points 1 and 2 (any new section on epigenetic mechanisms, authophagy, apoptosis, etc). There are limited changes that I found to be included in the revised V2 manuscript, and the changes were not marked. I suggest a highlighted text, different color of the text, or track change for all revised sections.”

A: We apologize for presenting a file without active “track-changes”. However, we need to clarify that the changes are not limited, in comparison with the first version of the manuscript. In fact the revision process was so extensive that we end up with a manuscript with numerous “track changes” annotations. Therefore, in order to facilitate the writing we presented a file in which “track changes” was not visible. We now realize that was not the best solution, and that the reviewer was not able to fully appreciate the corrections we made as a direct result of his/her comments. We regret that. To overcome that problem, in this second round of revisions we present the “track changes” highlighting all changes performed in the first and second round of revisions. In addition the reviewer can find in red the previous points raised (1 and 2). We also emphasize that we had already included the response to those points in the point-by-point response made in the first round of revisions.

  1. “A careful spelling and formatting check is required: i.e. Section 1.2 should read HOX genes and cancer not HOX gene and cancer. There are 39 HOX genes in mammals (not one HOX gene).”

A: We are grateful for the careful evaluation of our manuscript. We did an additional spelling and formatting check.

Reviewer 2 Report

  1. In point 2 of my first review, I mentioned that your manuscript mainly deals with mechanisms of HOX regulation which then impact invasion and metastasis. I see that you still keep this focus. To come to an end, I then ask you to change the heading of section 3 into "Five mechanisms of HOX deregulation affect invasion and metastasis of cancer cells".
  2.  Skip misleading lines 134 to 139. At this place this overview/retrospection is not helpful. 
  3.  Please differentiate between gene and protein. In line 70, the hoxasome is composed of HOX proteins and HOX protein cofactors.

Author Response

REVIEWER 2

  1. “In point 2 of my first review, I mentioned that your manuscript mainly deals with mechanisms of HOX regulation, which then impact invasion and metastasis. I see that you still keep this focus. To come to an end, I then ask you to change the heading of section 3 into "Five mechanisms of HOX deregulation affect invasion and metastasis of cancer cells."

A: We performed the correction suggested, not only in the heading of section 3 but also in the abstract and in point 1.2 of the introduction section.

  1. “Skip misleading lines 134 to 139. At this place this overview/retrospection is not helpful.”

A: We eliminated these lines.

  1. “Please differentiate between gene and protein. In line 70, the hoxasome is composed of HOX proteins and HOX protein cofactors.”

A: We took this in consideration in this second round of revisions and corrected the inconsistences. The gene names are differentiated in italic and the protein names are written in non-italic letters.